# Is Soil Bonitet an Adequate Indicator for Agricultural Land Appraisal in Ukraine?

**Leonid Shumilo [1,2,*], Mykola Lavreniuk [3] , Sergii Skakun [1,4] and Nataliia Kussul [2,3]**

[1]   Department of Geographical Sciences, University of Maryland, College Park, MD 20742, USA; skakun@umd.edu
[2]   Department of Mathematical Modelling and Data Analysis, National Technical University of Ukraine 'Igor Sikorsky Kyiv Polytechnic Institute', 03056 Kyiv, Ukraine; nataliia.kussul@lll.kpi.ua
[3]   Department of Space Information Technologies and Systems, Space Research Institute NAS Ukraine & SSA Ukraine, 03680 Kyiv, Ukraine; lavreniuk@ikd.kiev.ua
[4]   NASA Goddard Space Flight Center Code 619, Greenbelt, MD 20771, USA
[*]   Correspondence: lshumilo@umd.edu

**Abstract:** Agriculture land appraisal analysis is an important component of the land market. This task is especially essential for Ukraine, which plans to lift the moratorium on land transactions and legalize farmland sales in 2021. Most post-Soviet countries adopted the notion of a soil bonitet—a quantitative score representing natural soil fertility. This score is also proposed in Ukraine to perform agricultural land appraisals. However, this is a static parameter and does not account for the dynamics of actual crop production on the agricultural lands. Moreover, the bonitet score is not crop-specific. Therefore, in this study, we use maps of bonitet based on the soil map and natural-agricultural districts of Ukraine and crop yields at the village scale to explore the relationships between bonitet values and actual crop production in Ukraine. We found that land appraisal is not correlated with the actual soil bonitet.

**Keywords:** bonitet; agricultural land appraisal; soil quality; yield assessment

## 1. Introduction

Since the breakup of the Soviet Union in 1991, Ukraine has been experienced major changes in land cover and land use (LCLUC) [1]. The major drivers for these changes have been continuous economical and policy changes as well as climate variability. Since approximately 70% of land in Ukraine is devoted to agriculture (cropland, pasture, meadow), the changes were especially profound in the agricultural sector. In the past 5–10 years, these changes were particularly magnified due to: (i) the military conflict in the Eastern Ukraine [2], (ii) the conversion to double-cropping due to temperature increase [3] and a sharp increase in the production of industrial crops [4], (iii) the continuous practice of burning agricultural fields, and (iv) the preparation of the policy to open the land market [5]. The latter is directed to lift the moratorium on land transactions and legalize farmland sales. By the World Bank's estimates [5], establishing a transparent and efficient land market would boost economic growth by an estimated 0.5 to 1.5 percent per year over a 5-year period, depending on the reform design and complementary policies.

Within the open land market, land appraisal becomes one of the most important variables. Specifically, for agricultural lands, variables influencing land productivity would increase or decrease the value of the land parcel. In order to assess the value of agricultural land, many post-Soviet countries adopted the concept of a soil bonitet score [6–8]. Soil bonitet (B) is a quantitative assessment of its natural fertility and is expressed as a score in the range from 0 to 100. It is a component in the land appraisal technique that is officially used in Ukraine.

Within this assessment, bonitet is the only variable that is related to potential land productivity. The bonitet strongly depends on the soil type and bio-chemical characteristics

of the soil. While bonitet represents a potential value of the land in terms of productivity, it does not directly relate to the real (factual) land productivity and potential economical revenues. The former depends on multiple factors, including crop type and seeds, agricultural practices, meteorological conditions, and soil properties. Previous studies utilising remote sensing-based and bio-physical models [9–12] addressed the issues related to crop yield assessment in Ukraine.

Therefore, there can be a gap between the potential productivity and the real productivity of agricultural lands, which could alter the land appraisal value. While the soil bonitet can be considered as an integrated value, crop yields represent factual values, though they depend not only on the land properties itself. Therefore, there is an open question of how soil bonitet in Ukraine relates to specific crop yields in terms of agricultural productivity and land appraisal. In order to perform such an analysis, one has to incorporate crop maps, so crop-dependent relationships can be explored, as well as yields and bonitet scores at a finer-resolution scale.

The aim of this study is to show the real relationships between the actual economical capacity of the agricultural land represented in the yield on the village level and the main indicator of economic capacity of fields—the bonitet. The use of the bonitet indicator is widely discussed in the Ukrainian agricultural sector, so we have to provide essential decision-making information about the adequacy of its economic representation to support the reformation of the Ukrainian agrarian sector and opening of land market.

## 2. Materials and Methods

The present study takes advantage of multi-year, satellite-derived crop maps developed in 2016–2018 (Figure 1) [13] and village-level crop yields and bonitet values derived from soil maps to explore the crop-specific relationships between bonitet scores and crop yields in Ukraine. The results would give an insight if incorporating a bonitet score only into the land appraisal would provide an adequate metric of agricultural land value. Some recent studies proposed to use for this purpose additional information, such as crop rotation history [14], that can be obtained with use of available products.

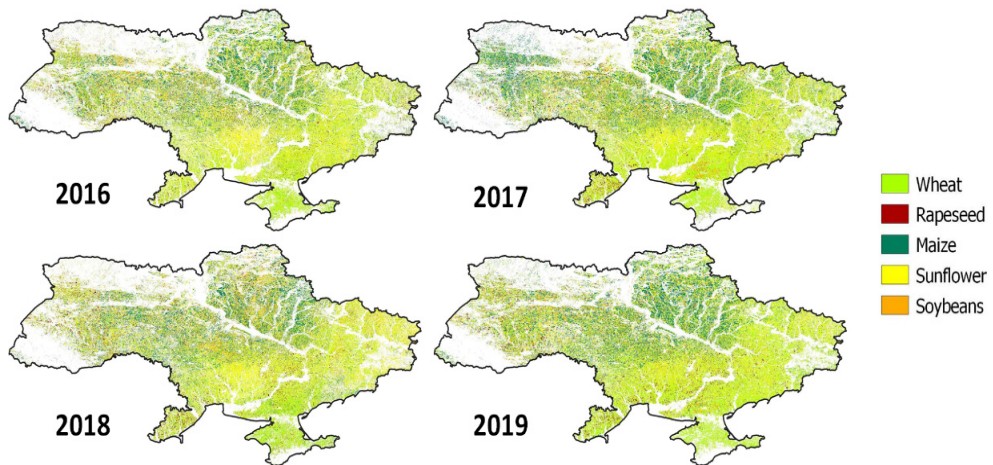

**Figure 1.** Crop type maps for Ukraine with majoritarian crop types for 2016–2018.

The analysis between crop yield and soil bonitet scores was performed using three major sources of data. The first one is official statistics on crop yields provided by the State Statistics Service of Ukraine for 2016–2018 at the village scale. The second data source is geospatial soil map for Ukraine, which is based on the national soil atlas data of Ukraine. The third source is natural-agricultural rayon's map. Using these maps with known bonitet values for each soil type and district, we generate a bonitet maps for Ukraine. Further analysis of these products was conducted using regression analysis and correlational analysis statistical techniques.



### 2.1. Yield Maps

Ukrainian governments provide yield statistics at the country and regional level. Since 2020, State Statistics Service of Ukraine provide statistics on the village level. This change provides new possibilities for yield assessment and analysis. The statistics are available for 2016–2018 years and cover all major crops: wheat, maize, soybean, and sunflower. These statistics available for all regions except Luhansk and Donets oblast that are under military conflict in the south of Ukraine and annexed by Russia Crimea. For further geospatial analysis, these data were aggregated using official village borders of Ukraine to obtain geospatial vector layers with the yield for each crop. Figure 2 shows an example of such geospatial layer for the sunflower yield statistics. The missing values are related to the areas without human settlements or cultivated fields by specific crop. The measurement unit for these statistics is the centner per hectare.

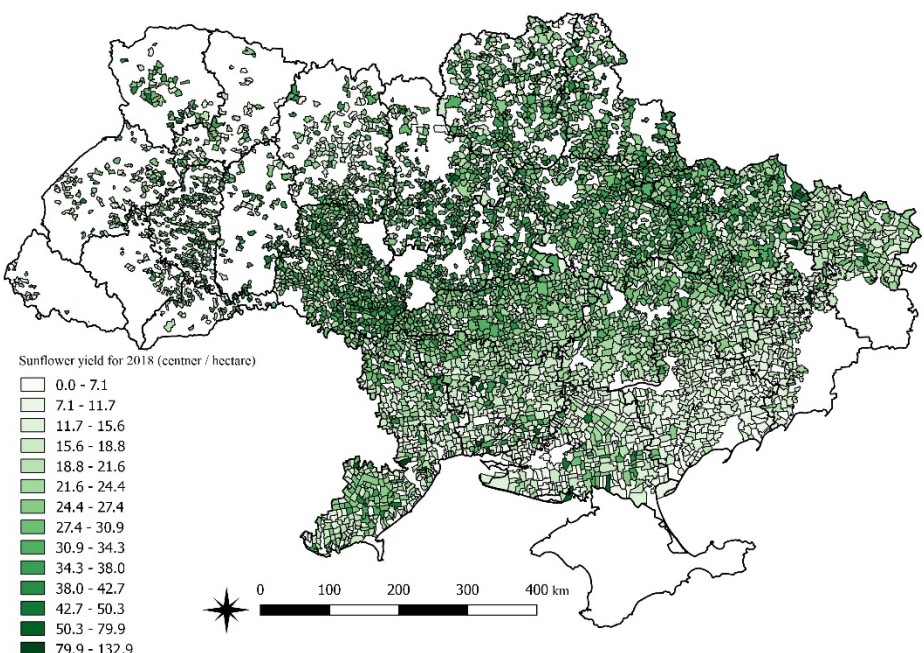

**Figure 2.** Geospatial layer with the sunflower yield statistics for 2018 aggregated with use of official village borders.

### 2.2. Bonitet Maps

Bonitet is the main soil quality characteristic used for the land appraisal. The estimation of field-level bonitet requires ground measurements and laboratory analysis of soil properties. As the result, not every field still has measured bonitet value and the aggregation of such data in the geospatial form for further analysis is not possible today, due to the limitation in the data availability. These limitations are related to the fact that as of now, bonitet data for Ukraine can be obtained only on the pages of official land auctions, they are not in digitized form. Due to the absence of convenient interface or instrument for the data access, it is not possible to automate the data collection process. In addition, for many fields, bonitet does not exist at all. Therefore, we used two sources of bonitet geospatial information. The first one is the official soil map of Ukraine, and the second one is a natural-agricultural map rayon's map of Ukraine. The second map is used as official reference data for estimation of $B_m$ coefficient in the state land appraisal laws of Ukraine.

To validate the obtained maps, we prepared ground truth data by digitizing bonitet documents for agricultural fields. The territory of Ukraine was covered by 79 ground samples for the validation.

### 2.2.1. Bonitet Map Production Based on the Soil Map

To produce soil bonitet map for Ukraine we assumed that soil bonitet depends on the soil type. Thus, the soil type map can be used as reference data for bonitet map. To produce the bonitet map shown in Figure 3, we assigned the standard soil type's bonitet values and corresponding soil types on the soil type map.

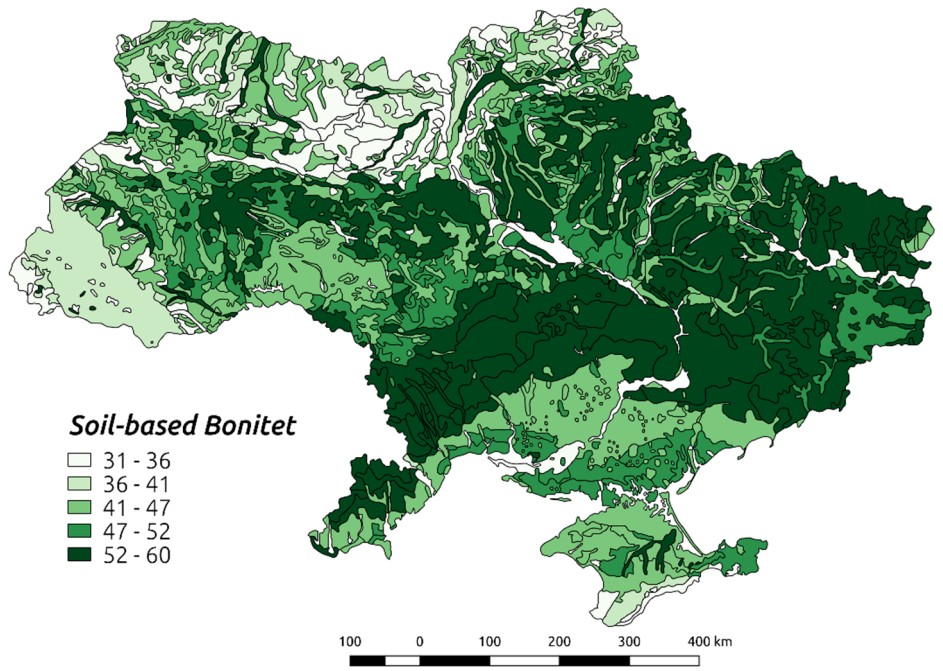

**Figure 3.** The bonitet map for Ukraine based on the soil atlas map.

This bonitet map contain 1200 units. The smallest unit has area of 349 th. ha, while the largest has area 2,721,723 th. ha. The average area of unit is 49,274 th. ha. Each unit has the bonitet value from 31 to 58. The average bonitet value for this map is 45.4, while the coefficient of variation is 15%. Therefore, this map has moderate variation of bonitet coefficient.

### 2.2.2. Bonitet Map Based on the Natural-Agricultural Rayon's Map

According to the Law of Ukraine "On Approval of the Methodology of Normative Monetary Valuation of Agricultural Lands", № 831, adopted by the Verkhovna Rada, formula (1) is used for agricultural land appraisal, where the coefficient $B_m$ is determined in the annexes for all natural-agricultural rayon's of Ukraine. This coefficient is considered to be the average bonitet score for each zone and was obtained during ground measurements. To conduct our experiments, we made a georeference and vectorization of natural-agricultural rayon's map of Ukraine to produce the bonitet map shown in Figure 4.

This bonitet map contain 205 units. The smallest unit has area of 8494 th. ha, while the largest has area 988,588 th. ha. The average area of unit is 291,481 th. ha. Each unit has the bonitet value from 10 to 71. The average bonitet value for this map is 37.28, while the coefficient of variation is 0.35. Therefore, this map has significant variation of bonitet coefficient.

### 2.3. Methods

In this study, we used two statistical methods to obtain analysis of dependences between yield and bonitet and to understand sufficiency of bonitet use in the land appraisal methodology. In addition, these methods were used to validate yield statistics data and soil-based bonitet map.

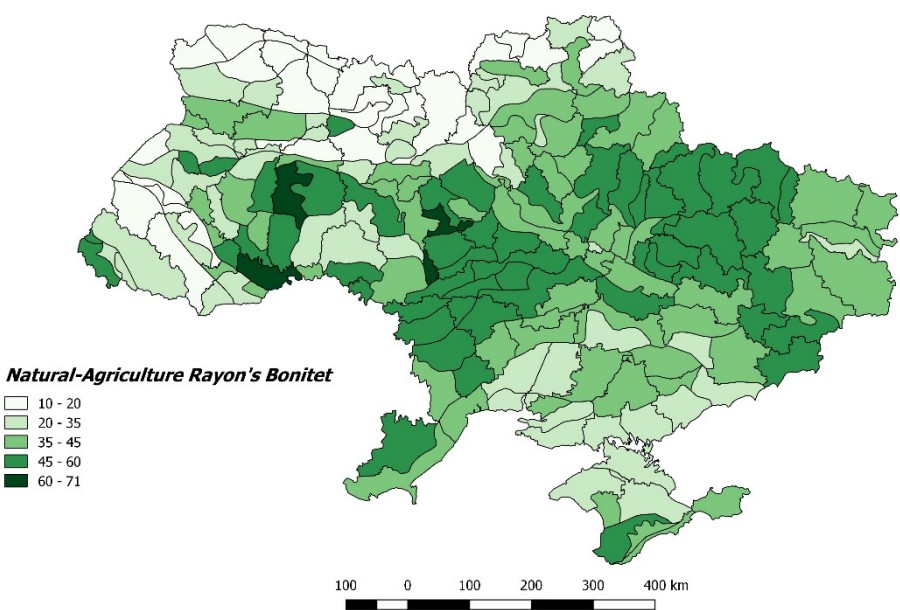

**Figure 4.** The bonitet map for Ukraine based on the natural-agricultural rayon's map.

### 2.3.1. Land Appraisal Methodology in Ukraine

Agricultural Land appraisal methodology in Ukraine consist in the use of multiple coefficients that reflect the distances between the field and some essential infrastructure objects or the location of the field relatively to the cities or recreational zones. The only economic variables that represent the quality of soil in terms of agricultural capacity or possible yield is the bonitet. The land appraisal in Ukraine can be represented by formula:

$$L_p = A * CRI * MC * \frac{B}{B_m},$$ (1)

where *A* is the area of agricultural land; *CRI* is the standardized capitalized rental income per unit area, which is determined by special tabulated tables; *MC* is the multiplication of coefficients, which considers location relatively to the settlements, resort and recreational areas, zones of radiation pollution, zonal factors of land location, and other coefficient related to the specification of land use and indexation of regulatory monetary valuation; *B* is the soil bonitet; and $B_m$ is the mean soil bonitet for the natural and agricultural area.

### 2.3.2. Regression Analysis

Regression analysis is the technique for data analysis that can be used for dependencies studies between two or more variables. It is common technique in the environmental and economy science. It based on the linear regression model, which was demonstrated in Equation (1), fitting with use of least square error method. Using this approach, it is possible to estimate $a_1$ and $a_0$ coefficients of the linear regression for further analysis:

$$Y = a_1 B + a_0, \text{ where :}$$ (2)

- *Y*—yield on the village level;
- *B*—mean village bonitet;
- $a_1$, $a_0$—regression coefficients, obtained by use of mean square error (MSE) method

The angular coefficient $a_1$ obtained by regression model fitting can be used for the hypothesis of the bonitet and yield dependence type selection. If the coefficient takes values very small or equal to 0, the hypothesis of lack of interdependence will be correct. In that case, when the modulus of the coefficient is much greater than 1, the interdependence is present, the function grows faster than the other. When the modulus of the coefficient is equal or close to 1, the model is close to the ideal and greatly describes the dependence

between these variables. In addition, the sign of the coefficient indicates if the relationship is direct or inverse.

### 2.3.3. Correlation Analysis

The second data analysis technique, which was used in this research, is the correlation analysis based on the Pearson coefficient. This statistical coefficient evaluates the linear dependence between two values and equal to multiplication of two vales covariations:

$$r(B,Y) = \frac{\sum_{i=1}^{m}(B_i - \overline{B})(Y_i - \overline{Y})}{\sum_{i=1}^{m}(B_i - \overline{B})\sum_{i=1}^{m}(Y_i - Y)}, \tag{3}$$

The correlation coefficient takes values between −1 and 1. If the coefficient takes value close to 0, the variables do not have any dependencies. If the absolute value is close to 1, the data dependence can be fully described using linear function. As well as for regression $a_1$ coefficient, the sign of the correlation coefficient indicates if the dependence is direct or inverse.

## 3. Results

### 3.1. Yield Data Validation

We have investigated the concordance of the crop yield data collected at the village council level for different years. For this purpose, we selected the yield data for each one of two years in a row and for each major crop. After this, we used regression and correlation analysis techniques to estimate the dependencies between yields for different years in the same villages. The results in terms of the visual dependency, the Pearson correlation coefficient, and two linear regression coefficients are shown in the Figure 5.

It is shown that the yield data for each crop has strong correlation for two years in a row (Pearson correlation coefficient is between 0.44 and 0.68). It means that there are no rapid or abrupt changes in the statistic data from year to year. The conclusion is that the statistical data are consistent and accurately collected. The differences and variations in yields for specific crop from year to year can be explained by different weather conditions (temperature and precipitation amount and distribution over the vegetation periods), mainly.

### 3.2. Impact of Bonitet Estimated from Soil Map on the Yield

For the first conducted experiment, we validated available soil-based bonitet map using collected ground-truth samples. The mean absolute error between ground-truth data and soil-based map is equal 8.39. To estimate more approximated accuracy, for each soil type, the mean ground truth bonitet was calculated. It was performed for comparison between ground truth bonitet and soil-based bonitet. In this case, the mean absolute error is equal to 6.32, and the Pearson coefficient is equal to 0.67.

To evaluate the impact of soil bonitet on the yield we explored dependence between these variables through the regression analysis and the correlation analysis. The results in terms of the visual dependency, the Pearson correlation coefficient, and two linear regression coefficients are shown in the Figure 6.

It is shown that the yield data for each crop do not have strong correlation with the soil bonitet index. In every case, we have found that the Pearson coefficients are very small and close to zero. So, we can conclude that there is no correlation between these values. Furthermore, the dependencies are also negative. Thus, we can assume that the soil-based bonitet value affects the fertility of the soil much less than other essential agricultural variables (agro-climatology, agricultural practices, and the history of crop rotations).

Such absence of dependencies between the yield data and soil bonitet shows that the soil quality indicator used for the agriculture land appraisal in Ukraine, namely, bonitet, does not meets with the land productivity capacity. So, other essential characteristics of location, such as agro-climatology, land use history, crop rotations, and agro-management

have much higher influence on the yield data. The field located in the area vulnerable to the droughts can have the same bonitet and price as the field in more sustainable areas for agriculture. Therefore, this bonitet information has no value for the land owner.

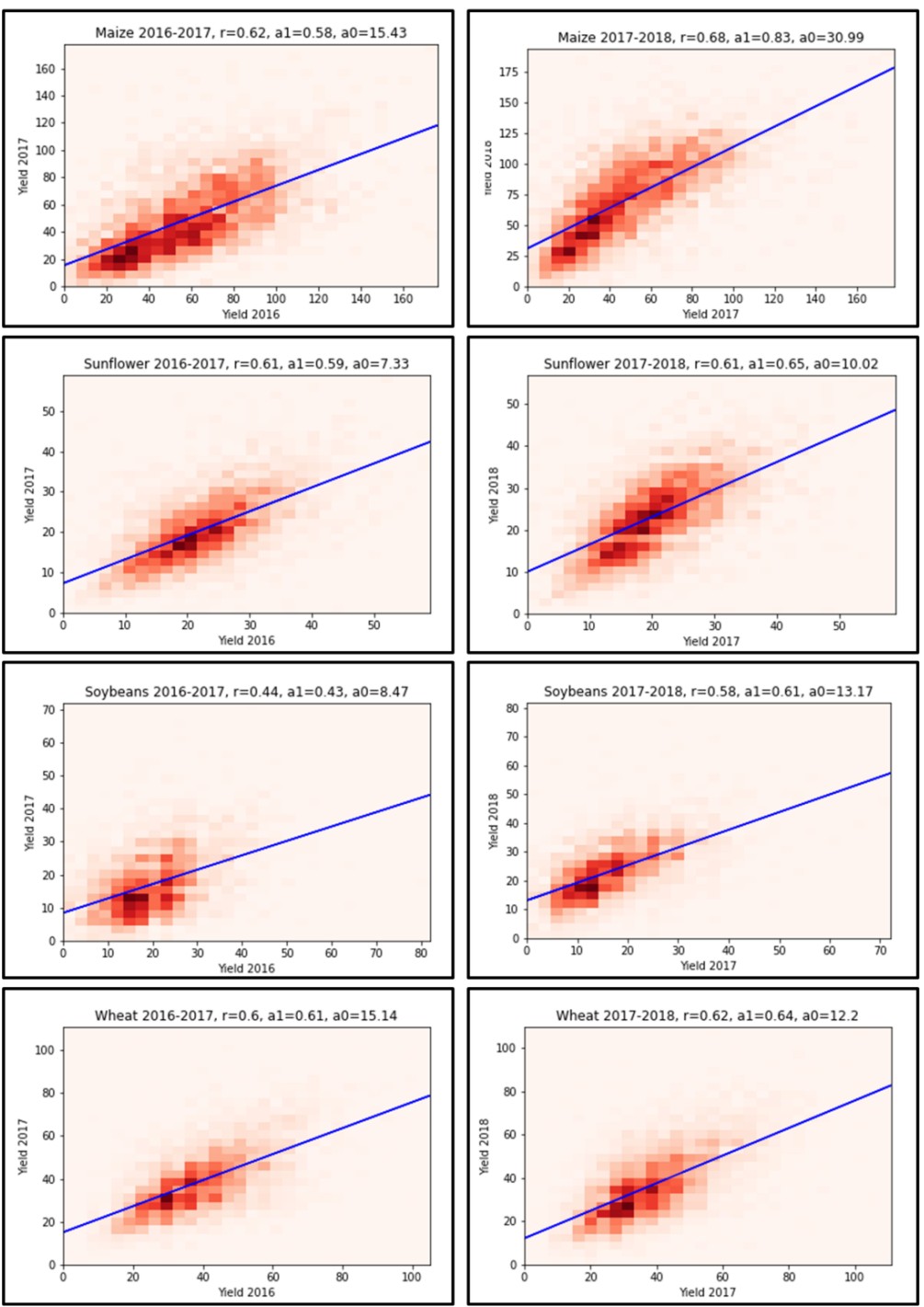

**Figure 5.** Dependency of the yield for the same crop and the same village councils within two consequent years. The background is a 2D histogram for two years yields, blue line is the regression function fitted on these values, *r* is Pearson coefficient, $a_1$ and $a_0$ are regression coefficients.

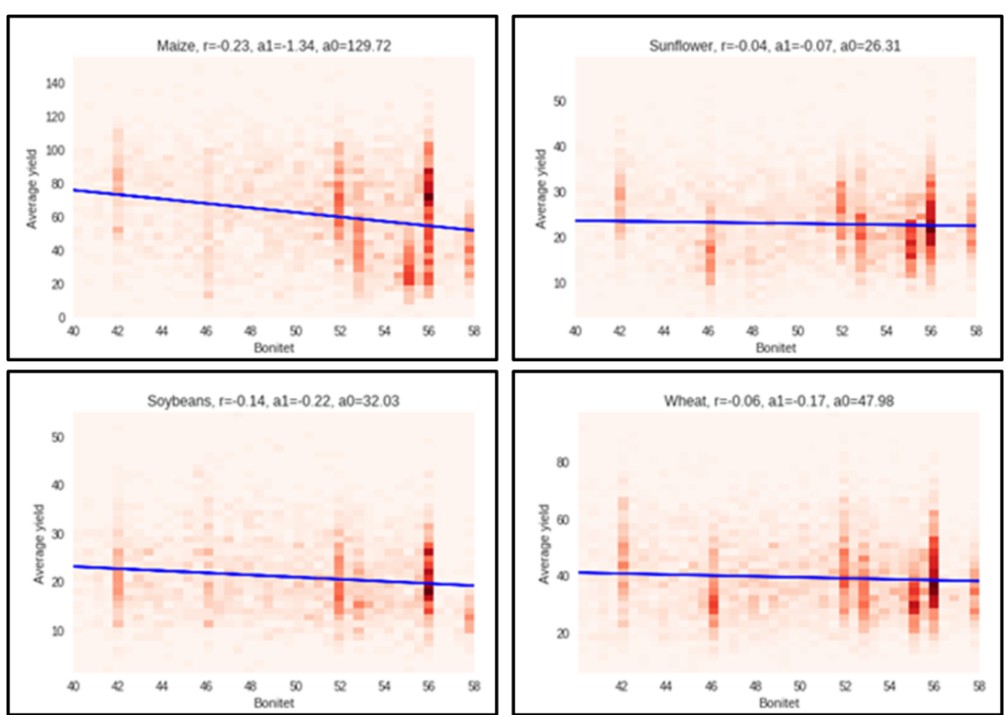

**Figure 6.** Dependence between average 3 years yield and soil-based bonitet on the village level. The background is 2d histogram for yield and bonitet values, blue line is the regression function fitted on these values, *r* is Pearson coefficient, $a_1$ and $a_0$ are regression coefficients.

### 3.3. Impact of Bonitet Estimated from the Natural-Agricultural Rayon's Map on the Yield

To conduct the second experiment, we validated the natural-agricultural rayon's based bonitet map using the same ground-truth samples. The mean absolute error between the ground-truth data and the soil-based map is equal to 10.33, and the Pearson coefficient equal to 0.36. Such a weak correlation and large error can be explained by the large average sizes of bonitet map units and significant variation coefficient. Thus, this map on the field level aggregation is not accurate. However, it still can be used for the village level analysis.

The experiment of the impact evaluation of natural-agricultural rayon's bonitet on the yield was conducted in the same way as at the 4.2 (Figure 7). The obtained results, as in the first case, indicate the absence of dependencies between bonitet and yield. Now, coefficients are small and two of four are negative. Thus, we can summarize that natural-agricultural zoning cannot be used as the reference data for land appraisal, because it cannot express the fertility of the soil. By the land appraisal logic, the $B_m$ value is used as a norm for the districts. It is an average bonitet value for it, so if the field bonitet in the district is higher than average, the land should have a higher price. In the opposite way, it works the same—if the land has lower bonitet, it should be cheaper. However, this experiment consumed that land with a higher bonitet value can have the same fertility as the land with a lower bonitet.

This experiment showed the same result as the previous one, so we have the same conclusion about the representation quality of the bonitet of natural-agricultural rayon's map as about the bonitet obtained from the soil map. However, if in the first case it was the approximate estimation of the bonitet values, now, it is the values that are used as $B_m$ in the official land appraisal methodology.

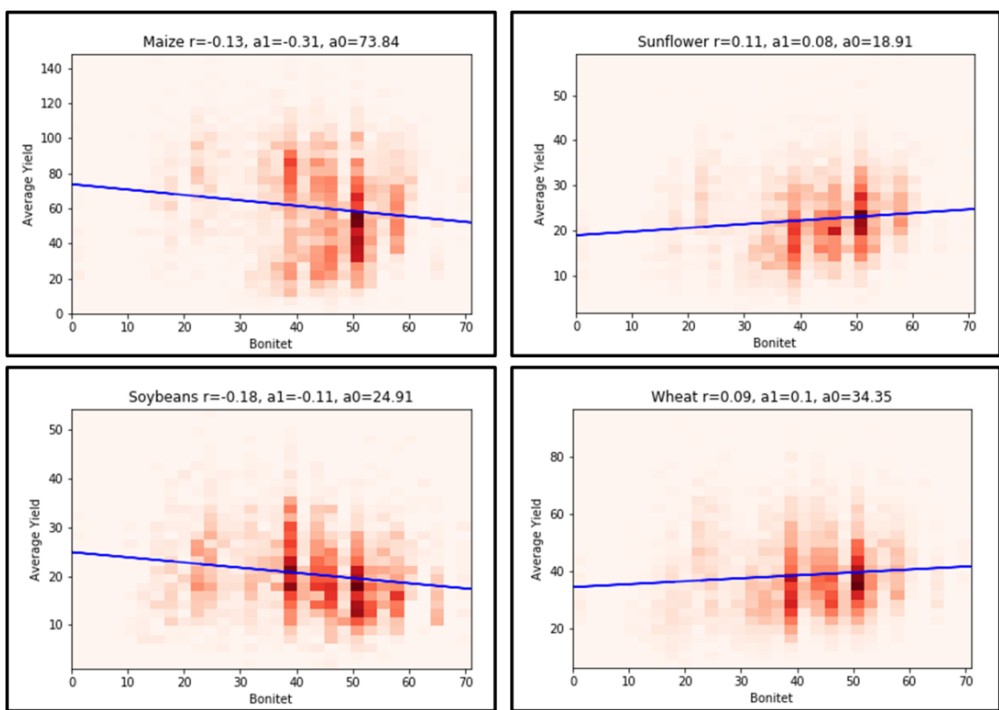

**Figure 7.** Dependency between average 3 year yield and natural-agricultural rayon's map bonitet on the village level. The background is 2D histogram for yield and bonitet values, blue line is the regression function fitted on these values, *r* is Pearson coefficient, $a_1$ and $a_0$ are regression coefficients.

## 4. Discussion

Using this obtained knowledge, Ukrainian policy makers could analyze how provided reforms for land market and in particular for land appraisal technique can be improved. The results provided by this research indicate that it is better to replace soil bonitet by more representative indicators. It is an important task because Ukraine is one of the biggest agri-food exporters in the Europe, and the land market opening will strongly affect Ukrainian economy. In addition, for the successful reform completion, laws governing the relations between landowners and land purchasers must comply with European standards.

The first way of land appraisal technique improvement may be the implementation of widely used land productivity sub-indicators. The historical biophysical information can be used to define if the land has signs of degradation or if it is sustainable or productive. In this case, when there are no available in situ information about historical biophysical characteristics, it is possible to use remote sensing data. The technology for the SDG 15.3.1 "Proportion of land that is degraded over total land area" assessment [15] can be very useful for this purpose. The main product used for the land degradation assessment is the land productivity map based on the vegetation indices' trends. Such a map can help to classify each land unit in one out of five classes: increasing productivity, stable not stressed, stable but stressed, early signs of decline, and declining productivity [16]. After, these classes can be used as the coefficients for the land appraisal. Instead of the use of static coefficients related to the field's location, this coefficient can be changed through the time by farmer's activities. These possible changes can stimulate the farmer to use more sustainable and environmentally friendly agrarian practices to increase the price of the land. From the other side, this coefficient can work as penalty for the use of unsustainable practices. In general, the implementation of the sustainable development goals' assessment practices can decrease the trend of desertification in the Ukraine and force agrarians to modernize their production by implementing more sustainable practices that can bring increase in the economy development [17]. Such an implementation of UN practices in Ukrainian agriculture is also consistent with the direction of Ukraine in the digitalization and the implementation of smart farming practices that can significantly boost the potential of

Ukrainian agricultural sector [18]. The only problem that makes it hard to implement such evaluation on the country level is the spatial resolution of satellite data. However, today's studies on the harmonization of Landsat-8 and Sentinel-2 [19] data are very promising, and today, it is already possible to build a land productivity map on the country level with 30 m spatial resolution [20].

The second way of land appraisal technique improvement may be the implementation of agro-climatic data. Ukrainian agrarians are still using old, out of date information about the agcro-climatic zoning of Ukraine, while the climatic conditions in the region has changed significantly in last decades. Such implementation requires the harmonization of official agro-climatic zoning and recent European research on the climatic zones [21]. The climatic variables are very important characteristics that can contain information about the probabilities of extreme weather events occurrence, such as droughts, floods, and high wind speed. Such events are very dangerous for the yield, so the agricultural land appraisal also should contain risk assessment component. Using these data, it is possible to obtain the crop yield potential of land [22] that can be used as an additional coefficient of land appraisal.

The last, but not least, possible way to improve the land appraisal technique in Ukraine is implementation of crop rotation history. The recent article of Deininger et al. [23] provide an analysis of dependence between crop yield and crop rotations on the village levels. This study shows that the violation of crop rotation rules has significant negative effect on the further yields, meanwhile, the use of good practices of crop rotations can provide sustainability of land or even slightly increase the productivity of land. The most common crop rotation violations in Ukraine are related to the plantation of technical crops that have negative effects on the land's biochemical characteristics, with higher frequencies then is allowed by the law [24]. As an example, sunflower planting is allowed only once per seven years. However, a lot of farmers are planting sunflowers every two or three years and sometimes even each year. The implementation of a crop history as a coefficient can solve many problems in Ukrainian agricultural regulation. If this coefficient will be based on the evaluation of the crop rotation history, then the effect can be similar to the implementation of SDG-based coefficient. It can decrease trends of desertification of land by stimulating farmers to use sustainable crop rotation schemes. In addition, this coefficient can act as a penalty for the farmers for the violation of crop rotation rules. After the implementation of this coefficient, all farmers are going to think twice when choosing between quick economic benefits that can be provided by monocropping and long-term benefits by saving or increasing land price, yield, and soil quality.

The further reformation and integration of more suitable indicators for the land appraisal should be completed to support the opening of Ukrainian land market. Right now, there is a lack of instruments to ensure the food security, land sustainability, and land appraisal probity. One of the main tasks of Ukrainian governances should be the protection of soils fertility and land productivity. The fines for the improper land use and crop rotation violation are too small and cannot avoid illegal actions that are leading to the land degradation. The absence of governmental control allows farmers to violate crop rotation rules and common agricultural practices. The benefits that can provide remote sensing information in the terms of land appraisal are related to the impossibility to hide or change information that is open and public. So, the satellite-based coefficients that are changing by the farmers actions are a good way to save the productivity of land. The violation of sustainable agricultural practices will bring long-term economic losses to the farmers, so they will be interested in the modernization of their food production.

## 5. Future Work

The further research of this problem requires significant changes in data availability policies. As it was mentioned in the article, bonitet data collection on the field level or village level require a lot of work, due to the data providers technical limitations. Thus, this procedure cannot be automated and require a lot of human work for data collection

and parsing. In addition, these data are available for a very limited number of fields and cover a very small territory. Therefore, we had to create geospatial bonitet layers based on the available for Ukraine soil map and natural-agricultural rayon's map. This approach is not so accurate, because the soil characteristics could be changed by years, while both maps are static for many years. This approach is the only way to currently conduct such research today.

## 6. Conclusions

Bonitet in Ukrainian law is an important characteristic of the soil quality that measure suitability of soils for growing crops. The agricultural land appraisal technique in Ukraine avoids the usage of important features, which are common for the crop yield assessment, such as agroclimatic conditions or agro-practices. The only variable that describes the yield potential of the agriculture land is the soil bonitet. In this article we analyzed how well this variable meets actual yield for croplands. This research is interesting from the geographical side because it shows a good example for the fusion of multi-scale maps and statistical data. From the economy side, it is also interesting, because the bonitet-based land appraisal technique is widely used in Eastern Europe and Asia, and it is the first analysis of bonitet metrics and yield dependencies on the village level. This research is unique because it was not possible to conduct bonitet sufficiency analysis for the full territory of Ukraine before, due to the bonitet and village-level statistical data availability problem. To conduct this research, we had to build geospatial bonitet maps based on the available soil map and natural-agricultural rayon's map for Ukraine. Based on these data, we calculated the mean bonitet values for each village and conducted correlational and regression analyses between village bonitet and yield for the majoritarian crops: wheat, maize, sunflower, and soybean. The analysis showed that the bonitet maps have very small correlation with actual yield. It means that bonitet does not meet the actual yield and food production potential for agricultural fields. Using these analytical results, policy makers in Ukraine have a scientific basis for the reforms implementation in the agriculture land market that meets European standards and have positive impact on the Ukraine economy. Ukrainian stakeholders and policy makers should consider more attention to the field's characteristics related to economic potential during land appraisal. At the same time, Ukrainian farmers and other land users should implement more sustainable agricultural practices to save the fertility of soil, to not lose the economic potential of owned land, and to meet the requirements of European agricultural policies. The main problem of this research is that field-level bonitet has much higher variability in comparison with the village level values. A more accurate analysis of the bonitet requires field-level bonitet data and yield data. However, due to the fact that correlation between village-level bonitet and village-level yield is very weak, we can assume that on the field level, it will not be much higher, especially when considering that other agro-management factors, such as fertilization or irrigation, have a very significant impact on the yield.

**Author Contributions:** Conceptualization, N.K. and S.S.; methodology, N.K., S.S., L.S. and M.L.; validation, L.S.; investigation, L.S. and M.L.; data curation, N.K., L.S., S.S. and M.L.; writing—original draft preparation, L.S., S.S., N.K. and M.L.; writing—review and editing, L.S.; visualization, L.S.; supervision, N.K. and S.S. All authors have read and agreed to the published version of the manuscript.

**Funding:** This research was funded within the NASA Land-Cover/Land-Use Change (LCLUC) Program, Grant Number 80NSSC21K0314. The authors acknowledge the funding received by the National Research Foundation of Ukraine from the state budget 2020/01.0273 "Intelligent models and methods for determining land degradation indicators based on satellite data" (NRFU Competition "Science for human security and society").

**Institutional Review Board Statement:** Not applicable.

**Informed Consent Statement:** Not applicable.

**Data Availability Statement:** Not applicable.

**Conflicts of Interest:** The authors declare no conflict of interest. The funders had no role in the design of the study; in the collection, analyses, or interpretation of data; in the writing of the manuscript, or in the decision to publish the results.

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
