# Peer review of "Is Soil Bonitet an Adequate Indicator for Agricultural Land Appraisal in Ukraine?"

_sustainability, doi:10.3390/su132112096_

Round 1

Reviewer 1 Report

Dear authors,

In this article results obtained show that the soil quality indicator used for the agriculture land appraisal in Ukraine, namely bonitet, is not meeting with the land productivity capacity. This study will help improve the system for assessing the fertility of Ukraine's lands, which will help improve the economic climate in this country going through difficult times. I have questions, but they are not of a fundamental nature. The article can be accepted for publication in this form. 

Best regards

Author Response

Dear reviewer! Thank you for high scoring of our article. During the revision we improved it by extending of discussion and conclusion sections.

Reviewer 2 Report

Dear authors

I have some comments to be considered to improve the manuscript.

  1. Introduction: I propose you move the equation and figure from that chapter to 3. Materials and methods and explain them in that chapter. You should explain the soil bonitet equation in the methods.
  2. Please state the clear aim of the study at the end of the Introduction.
  3. Please rename chapter 2 into = 2. Materials and methods
  4. Would you please join the Results and discussion chapter into 3? Results and discussion. Would you please discuss your results in each of the sub-chapters?
  5. Would you please extend the discussion chapter with soil bonitete impact on Ukrainian agricultural policy/strategy? Should it be further developed, and how this system relates to the international community/research? How is the soil protected, or how should it be. Please write at least 1 page of text. Good agricultural soil is a natural resource.
  6. Conclusions: Please answer the text to the following questions.

Why is this research unique?

What are the shortcomings/uncertainties of this research?

What did the scientific community learn out of it?

What are the benefits/recommendations for stakeholders (farmers, water managers)?

What are the recommendations for policymakers/legislators?

Future work?

Author Response

Dear reviewer! Thank you for your comments, there were very useful for our article. We consider all your comments and based on them made a new version of our article.

Brief response:

  1. Introduction: I propose you move the equation and figure from that chapter to 3. Materials and methods and explain them in that chapter. You should explain the soil bonitet equation in the methods.
    We moved the formula of land appraisal to the new sub-section “2.3.1. Land appraisal methodology in Ukraine” and the figure to the section 2 “Data, Materials and Methods”
  2. Please state the clear aim of the study at the end of the Introduction.

We added the clear aim of the study in the beginning of introduction,

  1. Please rename chapter 2 into = 2. Materials and methods

Done

  1. Would you please join the Results and discussion chapter into 3? Results and discussion. Would you please discuss your results in each of the sub-chapters?

Done

  1. Would you please extend the discussion chapter with soil bonitete impact on Ukrainian agricultural policy/strategy? Should it be further developed, and how this system relates to the international community/research? How is the soil protected, or how should it be. Please write at least 1 page of text. Good agricultural soil is a natural resource.

We extended the discussion section and tried to cover all this questions…

  1. Conclusions: Please answer the text to the following questions

We created new section – Future work and rewrite conclusion to address all mentioned questions.

Reviewer 3 Report

A rather good work, although there are some points in the text that they should be improved. In the attached pdf file you can see my comments or corrections needed.

Author Response

Dear reviewer! Thank you for a great review! We fixed all mistakes you mentioned in the review, extended discussion and conclusion and added clear aim of our research in the introduction.

Reviewer 4 Report

The authors aimed to answer the question how soil bonitet in Ukraine relates to specific crop yields in terms of agricultural productivity and land appraisal. The multi-year satellite-derived crop maps and village-level crop yields and bonitet values derived from soil maps to explore the crop-specific relationships between bonitet scores and crop yields in Ukraine were considered. As a result it was found that the bonitet maps have very small correlation with actual yield. It means that bonitet does not meet actual yield and food production potential for agricultural fields. However, the study results can be useful for policy makers in Ukraine, who obtained scientific basis for the reforms implementation in the agriculture land market that meets European standards and positively affect the Ukraine economy.

The manuscript clearly describes the presented study. However, neither the aim of the study nor the research hypothesis were clearly specified and both chapters Discussion and Conclusions are not correct and should be rewritten. In Discussion chapter there is no discussion with any other sources of literature. It includes rather summary and some conclusions, which should be specified in the next chapter. Conclusions chapter also includes some of a summary, some of the proper conclusions are on the end of the chapter.

Author Response

Dear reviewer! Thank you for your comments! We added the paragraph about aim of our research in introduction and rewrote, discussion and conclusions sections and made a few more edits to improve our article.